# CC Chemokines in a Tumor: A Review of Pro-Cancer and Anti-Cancer Properties of the Ligands of Receptors CCR1, CCR2, CCR3, and CCR4

**DOI:** 10.3390/ijms21218412

**Published:** 2020-11-09

**Authors:** Jan Korbecki, Klaudyna Kojder, Donata Simińska, Romuald Bohatyrewicz, Izabela Gutowska, Dariusz Chlubek, Irena Baranowska-Bosiacka

**Affiliations:** 1Department of Biochemistry and Medical Chemistry, Pomeranian Medical University in Szczecin, Powstańców Wielkopolskich 72, 70-111 Szczecin, Poland; jan.korbecki@onet.eu (J.K.); d.siminska391@gmail.com (D.S.); dchlubek@pum.edu.pl (D.C.); 2Department of Anaesthesiology and Intensive Care, Pomeranian Medical University in Szczecin, Unii Lubelskiej 1, 71-281 Szczecin, Poland; klaudynakojder@gmail.com (K.K.); Romuald.bohatyrewicz@pum.edu.pl (R.B.); 3Department of Medical Chemistry, Pomeranian Medical University in Szczecin, Powstańców Wlkp. 72, 70-111 Szczecin, Poland; izagut@poczta.onet.pl

**Keywords:** chemokine, CC chemokine, cancer, tumor, organ-specific metastasis, angiogenesis, lymphangiogenesis, tumor microenvironment, anti-cancer therapy, MCP-1

## Abstract

CC chemokines, a subfamily of 27 chemotactic cytokines, are a component of intercellular communication, which is crucial for the functioning of the tumor microenvironment. Although many individual chemokines have been well researched, there has been no comprehensive review presenting the role of all known human CC chemokines in the hallmarks of cancer, and this paper aims at filling this gap. The first part of this review discusses the importance of CCL1, CCL3, CCL4, CCL5, CCL18, CCL19, CCL20, CCL21, CCL25, CCL27, and CCL28 in cancer. Here, we discuss the significance of CCL2 (MCP-1), CCL7, CCL8, CCL11, CCL13, CCL14, CCL15, CCL16, CCL17, CCL22, CCL23, CCL24, and CCL26. The presentation of each chemokine includes its physiological function and then the role in tumor, including proliferation, drug resistance, migration, invasion, and organ-specific metastasis of tumor cells, as well as the effects on angiogenesis and lymphangiogenesis. We also discuss the effects of each CC chemokine on the recruitment of cancer-associated cells to the tumor niche (eosinophils, myeloid-derived suppressor cells (MDSC), tumor-associated macrophages (TAM), tumor-associated neutrophils (TAN), regulatory T cells (T_reg_)). On the other hand, we also present the anti-cancer properties of CC chemokines, consisting in the recruitment of tumor-infiltrating lymphocytes (TIL).

## 1. Introduction

A high percentage of deaths in cancer treatment is partly due to inadequate treatment methods as a result of our incomplete understanding of cancer mechanisms. However, cancer models and treatment methods are improving and increasing their efficacy. Over the last 20 years, the perception of cancer cells in tumors has changed significantly [1,2]. Previously, research focused on the inside of a cancer cell, but, nowadays, more attention is paid to the tumor niche and tumor microenvironment, with a special focus on non-cancer cells and intercellular communication [3,4,5]. It is now known that tumor growth, and, thus, the progression of cancer, requires communication between cancer and non-cancer cells, involving chemokines among other things.

Chemokines are a group of almost 50 chemoattractant cytokines and are divided into sub-families according to the domain found at the N-terminus [6,7]. One such sub-family are twenty-seven CC chemokines with an N-terminal CC domain, four of which (chemokine (C-C motif) ligand 6 (CCL6), CCL9/CCL10, and CCL12) are murine chemokines [6,7]. Although CC chemokines are presented using 28 symbols (CCL1 to CCL28), their actual number is 27, as chemokines CCL9 and CCL10 are actually the same chemokine. CC chemokines are ligands for ten classical receptors and serve as an important component of the tumor microenvironment and of cell relationships in the tumor niche, evidenced by their receptors association with the prognosis of patients with a particular type of tumor (Table 1 and Table 2 based on “The Human Protein Atlas” (https://www.proteinatlas.org/) [8,9]. There are no single CC chemokines that give the same prognosis in all types of tumors, which is due to the differences between cancers and the fact that any given chemokine has both pro-cancer and anti-cancer properties. A chemokine may cause tumor infiltration of the tumor by tumor-infiltrating lymphocytes (TIL), the cells that destroy cancer cells [10,11], whilst also recruiting tumor-associated cells that cooperate with cancer cells in tumor development [12,13].

As there is no comprehensive and up-to-date compendium that discusses the role of all human CC chemokines in cancer, the aim of this review was to collect all information about the involvement of each human CC chemokine in the hallmarks of cancer. Due to the large amount of data, we decided to divide the paper into two parts. In the first part we discuss CC chemokine receptors CCR1, CCR2, CCR3, and CCR4 together with their main ligands (CCL2, CCL7, CCL8, CCL11, CCL13, CCL14, CCL15, CCL16, CCL17, CCL22, CCL23, CCL24, and CCL26) (Table 3 and Table 4) [6,7]. In the second part, we present receptors CCR5, CCR6, CCR7, CCR8, CCR9 and CCR10 and their ligands (CCL1, CCL3, CCL4, CCL5, CCL18, CCL19, CCL20, CCL21, CCL25, CCL27, and CCL28) [14]. The described chemokines are presented according to the receptor that they activate. However, many CC chemokines are ligands for more than one receptor, and for this reason, we also discuss them according to their shared properties. In order to better understand the role of individual CC chemokines in cancer, special attention has been paid to their physiological functions.

## 2. CCR1

### 2.1. CCL14

CCL14 (also known as hemofiltrate CC chemokine (HCC)-1), occurs in high concentrations in plasma [15]. The expression of this chemokine has also been found in organs such as spleen, colon, small intestine, liver, muscle, and bone marrow but not in the brain and kidney [15]. CCL14 is a ligand of CCR1 [16,17], CCR5 [17,18], and a weak agonist of CCR3 [17]. For this reason, it serves as a chemotactic agent for monocytes but not for T lymphocytes, neutrophils, and eosinophils [15], although a later study showed that CCL14 may be a neutrophil chemoattractant [19].

CCL14 is secreted as an inactive proform. Following the cleavage of the 8-amino acid fragment by the urokinase-type plasminogen activator (uPA), plasmin or kallikrein-related peptidases, it is converted to an active form [20,21,22,23], which may be further processed by the cleavage of two amino acids by dipeptidyl peptidase IV (DPPIV)/CD26, which inactivates the active form of CCL14 [24]. The active form of CCL14 can also be broken down by atypical chemokine receptor 2 (ACKR2)/D6 [24]; however, the expression of this receptor decreases as the tumor develops [25,26,27]. 

The physiological function of CCL14 is poorly understood. It is important in the functioning of bone marrow by stimulating the proliferation of CD34^+^ human bone marrow cells [15]. CCL14 causes changes in extracellular matrix in the trophoblast, which induces trophoblast cell migration and embryo implantation [28].

CCL14 has anti- and pro-cancer properties. The expression of this chemokine is reduced in many solid tumors including in liver, breast, lung, and prostate cancer [29,30]. On the other hand, it is elevated in brain and esophageal cancer [30]. Since CCL14 is present in high concentrations in plasma and is activated by appropriate proteinases [15,20,21], it can act locally at the site of proteinase activity, e.g., uPA and plasmin. 

CCL14 reduces the activation of the Wnt/β-catenin pathway in hepatocellular carcinoma cells, which inhibits their proliferation and causes their apoptosis [29]. CCL14 also has pro-cancer properties—it induces the migration of cancer cells, as shown by an experiment on breast cancer cells [31], and also causes angiogenesis [31]. It is also postulated that the CCL14→CCR1 axis is crucial in liver metastasis [15,32]. In addition, CCL14 participates in the recruitment of monocytes into the tumor niche, especially to bone marrow, as shown on a multiple myeloma model [33]. The monocytes are then converted into tumor-associated macrophages (TAM) by the tumor microenvironment. Subsequently, CCL14 increases the proliferation of TAM, which increases the number of these cells in the tumor niche [33]. Although CCL14 can participate in the infiltration of the tumor by anti-cancer TIL [30], a study on hepatocellular carcinoma has shown a negative correlation between the concentration of CCL14 and such infiltration [30], and so the effect of CCL14 on the response of the immune system to cancer requires further study.

### 2.2. CCL15

CCL15 (also known as HCC-2, macrophage inflammatory protein (MIP)-1δ, MIP-5, leukotactin-1) activates two receptors: CCR1 [34,35,36] and CCR3 [34,35], with the N-terminally truncated form of this chemokine showing a strong affinity for CCR1 [22,37,38]. CCL15 is a chemotactic agent for monocytes, eosinophils, and neutrophils [35,36]. It has the greatest expression in the gut and liver [36] and so is crucial for maintaining immune balance in these organs. 

CCL15 is a serum biomarker and independent predictor of survival in hepatocellular carcinoma [39,40]. The higher the concentration of CCL15, the worse the prognosis—an effect associated with the migration and invasion of these cells, induced by CCL15 via CCR1 [40]. In head and neck squamous cell carcinoma, activation of CCR1 by the described chemokine induces apoptosis resistance and drug resistance by activating nuclear factor κB (NF-κB) [41]. CCL15 is crucial for the metastasis of renal cell carcinoma. If the cancer cell stops in the bone, CCL15, produced by the cancer cell, acts chemotactically on osteoclast precursors and osteoclasts, probably via CCR1 and CCR3 [42]. In this case, CCL15 causes osteoclastogenesis, which is followed by bone remodeling around the tumor cell and the formation of a metastatic niche [43]. The CCL15→CCR1 axis is also postulated to play an important role in liver metastasis [32,36]. 

CCL15 also acts on non-cancer cells in the tumor, for example causing angiogenesis mediated by CCR1 and CCR3 on vascular endothelial cells [44]. In addition, in hepatocellular carcinoma and colorectal cancer, CCL15 is responsible for recruiting TAM and myeloid-derived suppressor cells (MDSC) [45,46] and for recruiting mesenchymal stem cells (MSC) into the tumor niche [47]. Colorectal cancer models have shown that CCL15 causes tumor-associated neutrophils (TAN) recruitment to the tumor niche via CCR1 on these cells [48].

### 2.3. CCL16

Chemokine CCL16 (other names: HCC-4, liver expressed chemokine (LEC), liver-specific CC chemokine-1 (LCC-1)) is constitutively expressed in the liver and by hepatoma cells [6,49], as well as by monocytes treated with interleukin (IL)-10 [50]. High concentrations of CCL16 can be found in the blood [51]. It is a ligand for CCR1 [51,52,53,54], CCR2 [51], CCR5 [51] and CCR8 [52]. CCL16 is a ligand of the histamine H4 receptor and therefore may act on eosinophils [55]. 

Receptor CCR1 is important in liver metastasis [32] and so it can be postulated that CCL14 [15], CCL15 [36] and CCL16 [49], i.e., chemokines with a high expression in the liver and at the same time, ligands for CCR1, may cause liver metastasis. Another pro-cancer property of CCL16 is the induction of angiogenesis, related to the expression of its receptor CCR1 on vascular endothelial cells [53]. CCL16 also causes the migration of cancer cells if they exhibit CCR1 expression [54]. 

On the other hand, CCL16 enhances the anti-cancer effects of cytotoxic T and dendritic cells (DC) lymphocytes [56,57] and some consider the possibility of using of this chemokine for enhancing the anticancer response in cancer immunotherapy [58,59,60,61].

### 2.4. CCL23

CCL23 (also known as CKβ8, MIP-3, and myeloid progenitor inhibitory factor-1 (MPIF-1)) is produced by eosinophils [62], monocyte-derived dendritic cells [63] and monocytes activated by interleukin (IL)-1β [64]. It serves as a chemoattractant for dendritic cells, resting T lymphocytes and monocytes, but not in T cells and eosinophils [63,64,65,66,67]. CCL23 is a ligand for CCR1 [67,68], and the N-terminally truncated form of this chemokine has a strong affinity for CCR1 [37,38]. Due to alternative splicing, CCL23 occurs in two forms, shorter CCL23α/CKβ8 and longer CCL23β/CKβ8-1 [66,67]. The shorter form activates CCR1 more strongly [66]. However, CCL23β can be cleaved at the C-terminus [69], which results in the release of SHAAGtide, a peptide which is a ligand for formyl peptide receptor-like 1 (FPRL1)—the activation of this receptor causes the migration of monocytes and neutrophils. CCL23 suppresses differentiation of myeloid progenitor cells [70]. For this reason, increased expression of CCL23 in acute myeloid leukemia cells leads to the suppression of hematopoiesis [71]. 

There are few studies indicating the involvement of CCL23 in cancer. However, it seems that this chemokine has both pro- and anti-cancer properties. It stimulates the proliferation of cells with CCR1 expression [72]. For this reason, it can be assumed that CCL23 increases the proliferation of cancer cells. CCL23 is secreted by macrophages in omentum and so the CCL23→CCR1 axis is crucial for omentum metastasis in ovarian cancer [73]. CCL23 is also a chemotactic agent for osteoclast precursors via CCR1 on these cells [74] and so it may play some role in bone remodeling and the formation of a metastatic niche in bones. However, the most important pro-cancer function of CCL23 may be the induction of angiogenesis by activating CCR1 on vascular endothelial cells [75,76], a process associated with increased expression and secretion of matrix metalloproteinase (MMP)-2 from these cells. CCL23 also induces an increase in the expression of the kinase insert domain-containing receptor (KDR)/fms-like tyrosine kinase 1 (Flk-1) on vascular endothelial cells [77], which enhances the effect of vascular endothelial growth factor (VEGF) on these cells. However, there are no studies on the involvement of CCL23 in angiogenesis inside the tumor. 

CCL23 can also influence cancer-associated cells. For example, this chemokine may contribute to the function of eosinophils in a tumor [62]. It is also a chemoattractant for various cells of the immune system and therefore may cause the recruitment of some cells into the tumor niche [64,65,66,67,68]. It is yet to be established whether it causes tumor infiltration by anti-cancer TIL or participates in the recruitment of cancer-related cells, in particular MDSC, TAM, TAN and regulatory T cells (T_reg_).

## 3. CCR2 and Its Ligands: CCL2, CCL7, CCL8, and CCL13

CCL2 (also known as monocyte chemoattractant protein (MCP)-1) [78], CCL7 (also known as MCP-3) [79,80], CCL8 (also known as MCP-2) [81] and CCL13 (also known as MCP-4) [80,82,83] are ligands for CCR2. However, CCL2 can also activate CCR4 [84], CCR5 [85] and—at high concentrations—CCR1 [86]. CCL2 is also an antagonist of CCR3 [87]. CCL7 can also activate CCR1 [79,88] and CCR3 [89] and is an antagonist of CCR5 [85]. CCL8 can also activate CCR1 [81], CCR3 [89] and CCR5 [85,90,91]. CCL13 is also a ligand for CCR3 [82,83,89] and CCR5 [85]. CCL2, CCL7, and CCL8 are proteolytically spliced at the C-terminus, which is necessary for the acquisition of chemotactic properties by these chemokines [22]. However, cleavage at the N-terminus by MMPs makes them antagonists of their own receptors [22,92]. Another mechanism for reducing the activity of the described subgroup of chemokines is the chemokine decoy receptor ACKR2/D6 which reduces the level of this and many other CC chemokines [25,26,27]. However, the expression of this receptor in tumors is gradually reduced along with the progress of tumor growth.

One of the most important functions of the discussed subgroup of chemokines is the recruitment of monocytes to inflammatory reaction sites [93,94,95,96]. However, the discussed chemokines are involved in the recruitment of basophils, T cells, and NK (natural killer) cells [93,97]. In addition, CCL7 and CCL8 are chemoattractants for eosinophils [93]. Due to the activation of CCR3 by CCL13, this chemokine is important in the pathogenesis of allergic inflammation and asthma because it shows chemotactic activity against monocytes and eosinophils [82,98,99]. Due to the recruitment of monocytes, CCL2, CCL7, CCL8, and CCL13 are important in the pathogenesis of many diseases where an important role is played by monocytes and macrophages, in particular atherosclerosis, inflammatory bowel disease, and cancer. 

The expression of chemokines from this subgroup increases in many cancers. Breast cancer is associated with increased CCL7 expression [100], while glioblastoma multiforme is accompanied by increased concentrations of CCL2 and CCL7 [101]. On the other hand, in some cancers, e.g., in ovarian adenocarcinoma, the expression of CCL2 decreases [102]. 

In a tumor, CCL2 is produced by cancer cells (Figure 1) [103,104,105,106,107,108,109]. The expression of this chemokine may be increased by factors such as growth factors [110], radiotherapy [107], cycling hypoxia [111] and anti-cancer drugs [112], or interactions with other cells, including cancer-associated fibroblasts (CAF) [113]. However, the expression of this chemokine in a tumor also occurs in MDSC [114], MSC [115], TAM [104,116,117,118], TAN [119], and CAF [120,121,122]. CCL7 and CCL8 are also expressed in CAF [123,124,125] and TAM [126]. The expression of CCL2, CCL7, and CCL8 in CAF is very important in cancer. Their expression is increased in CAF under the influence of interaction with a cancer cell, an important effect in the initial stages of metastatic niche formation and the functioning of a tumor [123,124,127,128,129,130]. 

Increased expression of CCL2 in a tumor is associated with a worse prognosis for patients with solid tumors [131]. This is associated with the multiple pro-cancer properties of this chemokine. The most important function of CCL2 is the CCR2-mediated recruitment of TAM [12,13,103,132,133,134,135] and MDSC [13,122] into the tumor niche. CCL2 is also one of the factors contributing to M2-type macrophage polarization [136]. It can also recruit one of the T_reg_ subsets that shows CCR2 expression [137,138]. The recruitment of other T_reg_ subsets by CCL2 may also depend on the activation of CCR4 by this chemokine [84]. CCL2 can also recruit T helper type 17 (Th17) [139] and MSC [140] into the tumor niche. In brain tumors, CCL2 participates in the recruitment of neural progenitor cells [141] and microglia [142]. In liver cancer, CCL2 additionally causes the recruitment of hepatic stellate cells [143]. TAM recruitment can also be induced by CCL7 [144] and CCL8 [145], while T_reg_ are recruited into the tumor niche by CCL8 [91] into the tumor niche. 

CCL2 [10,11,146,147,148] and CCL7 [149,150] may also cause infiltration of the tumor by TIL, which has an anti-cancer effect. However, it seems that CCL2 in a tumor interferes with the function of anti-cancer T lymphocytes [151] and dendritic cells [152]. The discussed group of CC chemokines (CCL2, CCL7, CCL8, and CCL13) have numerous pro-cancer functions, which outweigh their anti-cancer properties. For example, the prognosis for patients is worse when the expression of CCL2 in the tumor is elevated [131]. 

CCL2 and CCL8 increase cancer cell proliferation [108,153,154]. CCL2 and CCL8 also cause enhanced tumor cell stemness and cancer stem cells self-renewal [127,155]. CCL2 also increases apoptosis resistance and drug resistance by activating the phosphatidylinositol-4,5-bisphosphate 3-kinase (PI3K)→ Akt/protein kinase B (PKB)→ mammalian target of rapamycin (mTOR) pathway [156,157,158]. This is of great importance as the expression of this chemokine can be increased by radiotherapy [107] and anti-cancer drugs [112]. For this reason, it is postulated to administer additional drugs to disrupt the function of CCL2 during cancer therapy. 

CCL2 [86,156,159,160], CCL7 [123,161], and CCL8 [124,125,159] cause cancer cell migration. CCL2 also causes epithelial-to-mesenchymal transition (EMT) by activating the extracellular signal-regulated kinase (ERK) mitogen-activated protein kinase (MAPK)→ glycogen synthase kinase-3β (GSK-3β)→ Snail and PI3K→ Akt/PKB pathways [162,163]. CCL7 [164] and CCL8 [155,165] also cause EMT. CCL2 is crucial for the subsequent stages of metastasis. After EMT, a cancer cell increases the expression of CCL2 [166]. This allows it to effectively recruit macrophages, which participate in the early stage of the development of a metastatic niche [134,167]. However, at this stage CCL2 can also recruit neutrophils, which have a destructive effect on cancer cells [168]. 

In further stages of metastasis, CCL2 participates in the formation of a metastatic niche. As CCL2 is secreted by bone marrow endothelial cells [169], it causes the diapedesis of cancer cells into the bone tissue. Bone metastasis in prostate cancer is closely related to osteoblasts because prostate cancer cells secrete parathyroid hormone related proteins (PTHrP) that cause an increase in CCL2 expression in osteoblasts [170,171]. CCL2 can also be directly produced by other cancer cells [106,172,173]. Then the described chemokine causes the differentiation of osteoclasts, which are involved in bone remodeling and the formation of a metastatic niche [174]. CCL2 is also important in the perineural invasion of cervical cancer and prostate cancer [175,176], due to the expression of CCL2 in the nervous tissue. In particular, this chemokine is secreted by Schwann cells and supports the formation of metastasis by acting on cancer cells. 

CCL2 can also directly cause angiogenesis, by acting on vascular endothelial cells on which it is expressed CCR2 [104,177]. However, some studies have shown that vascular endothelial cells in tumors do not express CCR2 [103,133], and so it seems that CCL2 may indirectly cause angiogenesis by recruiting TAM and increasing VEGF-A expression in these cells [132,133,135,178,179]. Angiogenesis may also be indirectly caused by the CCL2-mediated increase in VEGF expression in a cancer cell [108,156]. In comparison, there are no published studies showing the direct effects of CCL2, CCL7, CCL8, and CCL13 on lymphangiogenesis. It is possible that these CC chemokines have an indirect impact on this process via tumor-recruited TAM [180]. 

There are no studies showing the importance of CCL13 in cancer. However, it has been proven that this chemokine may increase apoptosis resistance [181]. In a tumor, it might cause cancer cells to become drug resistant. As this chemokine activates CCR2 and CCR3, it should have the same properties as other ligands for these receptors: CCL2 and eotaxins, but this needs to be confirmed by further research.

## 4. CCR3 and Eotaxins: CCL11 (Eotaxin-1), CCL24 (Eotaxin-2), and CCL26 (Eotaxin-3)

Eotaxins are three CC chemokines: CCL11 (eotaxin-1), CCL24 (eotaxin-2, also known as CK6 and MPIF-2) and CCL26 (eotaxin-3). Their most important receptor is CCR3 [182,183,184,185]. CCL11 is also a ligand of CCR5 [85,186] but also an antagonist of CCR2 [186]. CCL24 is an antagonist of CCR2 [87]. CCL26 is a ligand of CX3C motif chemokine receptor 1 (CX3CR1) [187,188] but also an antagonist of CCR1, CCR2, and CCR5 [189,190]. Eotaxins are the main chemotactic factors for eosinophils and to a lesser extent for basophils [7,183,191]. For this reason, eotaxins play a significant role in the pathophysiology of allergic reactions [184,192,193,194].

An elevated expression of eotaxins also occurs in tumors such as breast cancer [100], colorectal cancer [195] and oral squamous cell carcinomas [196], where it is associated with the recruitment of eosinophils into the tumor niche (Figure 2). However, the role of eosinophils in tumor is not clear [197,198,199], as they show both pro- and anti-cancer characteristics depending on the type of tumor. In addition to the recruitment of eosinophils, CCL26 has also been shown to cause TAM recruitment depending on CCR3 [200] and CX3CR1-dependent recruitment of MDSC into the tumor niche [188].

Eotaxins do not only influence the composition of cells in the tumor niche. Increased expression of CCR3 has been reported in tumors such as renal cell carcinoma [201] or glioma [101]. Activation of this receptor on a cancer cell increases proliferation and migration [201,202,203]. CCL11 causes cancer cell apoptosis resistance by activating ERK MAPK [204]. Eotaxins also increase tumor vascularization. Due to the fact that CCR3 is expressed on vascular endothelial cells, eotaxins—especially CCL11—cause angiogenesis [205,206]. CCL11 may also indirectly affect angiogenesis, as the activation of the CCR3 receptor by CCL11 increases VEGF expression in the hepatocellular carcinoma cells and thus promotes angiogenesis [207]. On the other hand, angiogenesis is inhibited by eosinophils recruited by eotaxins, which leads to the necrosis of some areas in a tumor [199].

## 5. CCR4 and Its Ligands CCL17 and CCL22

CCL17 (also known as thymus and activation regulated chemokine (TARC)) [208] and CCL22 (also known as macrophage-derived chemokine (MDC)) [209] are the ligands for CCR4. These chemokines are the chemotactic factor for Th2 and T_reg_ due to the expression of the CCR4 on these cells [210] and for this reason they are important in the pathogenesis of asthma and allergy. CCL17 and CCL22 are also crucial in the homing of lymphocytes to the skin [210]. They exert an anti-cancer effect by causing the infiltration of TIL into the tumor [211,212], a process dependent on CCR4 on these cells.

On the other hand, the expression of CCL17 and CCL22 is elevated in a breast cancer tumor [100], CCL22 expression is increased in colorectal adenocarcinomas [213], while CCL17 expression is increased in glioblastoma multiforme [101]. CCL17 and CCL22 are both produced by cancer cells in a tumor (Figure 3) [214,215] and by TAM [118,214,216,217,218,219]. CCL17 is also produced in TAN [119,220] and CAF [221].

CCL17 and CCL22 are responsible for CCR4-dependent recruitment of T_reg_ into the tumor niche, which enhances cancer immune evasion [220,222,223,224,225]. They can also recruit eosinophils [226] and Th17 [227] into the tumor niche. At the same time, these chemokines may also cause infiltration of the tumor by anti-cancer TIL [211,212]. Due to the recruitment of cancer-related cells in the squamous cell carcinoma of the tongue into the tumor niche, increased expression of CCL22 is associated with a poorer prognosis [219]. The same applies to the increased expression of CCL17 in breast cancer [228]. In contrast, an increased expression of CCL22 in the tumor improves prognosis in lung cancer patients [229] and elevated CCL17 blood levels improve prognosis in melanoma patients [230].

Although are no studies showing the direct effects of CCL17 and CCL22 on angiogenesis, T_reg_ recruited by CCL17 and CCL22 do have pro-angiogenic properties via the secretion of VEGF [231].

CCL17 and CCL22 promote stemness of cancer cells with CCR4 expression [218], drug resistance [232], stimulate the proliferation of cancer cells [233], and cause cancer cell migration and EMT, as shown on many types of cancers [214,218,234]. Cancer cell migration is associated with metastasis. CCR4 expression has been linked to lymph node metastasis [214] and omental milky spots metastasis [235]. The CCL22→CCR4 axis also participates in bone metastasis due to the high expression of CCL22 in bones [236]. Brain [237] and lung [238] metastasis have also been associated with CCR4 expression. However, in these organs the expression of CCR4 ligands is low [239] and for this reason, the described axis may participate in metastasis only at the stage of induction where there is cell migration from the parent tumor.

## 6. CC Chemokines in Therapy

Many of the aforementioned CC chemokines simultaneously cause infiltration of a tumor by anti-cancer TIL while also recruiting cells which support the growth of the tumor. As the action of these chemokines depends on the tumor microenvironment [240], this a key area that should be studied in order to develop more effective therapeutic approaches. Importantly, a balance between pro- and anti-cancer mechanisms in the tumor microenvironment differs depending on the type of cancer, which means that an increased expression of a given CC chemokine may either improve or worsen a prognosis, as shown in Table 1 above [8,9]. Despite this, cancer therapies which target CC chemokines hold a lot of promise, as shown by in vivo and clinical trials, especially in combination with immunotherapy [241,242,243,244].

To date, few studies have centered on the significance of the ligands of CCR1 as therapeutic targets in cancer therapies. The most researched ligand has been CCL16 whose increased expression has shown an anti-cancer effect in mouse breast cancer [60,245], and colon carcinoma [245], and prostate cancer [246] due to an increase in the infiltration of the tumor by CD4+ T cells, CD8+ T cells, and DC. This effect may be enhanced by the use of factors that intensify the immune response, such as CpG (Toll-like receptor 9 ligand) and anti-IL-10 receptor antibody [245,246]. Another therapeutic approach includes the use of a CCR1 antagonist, for example BL5923, which reduces the recruitment of immature myeloid cells into the tumor and inhibits liver metastasis in mice with colon cancer [247].

CCL2 and CLL7 have also been well researched as therapeutic targets in cancer therapy due to their significant function in tumors. An increased expression of these ligands is often associated with a worse prognosis in various cancers [8,9,131]. Therefore, the use of CCL2 siRNA, CCR2 siRNA, CCL2-neutralizing antibodies, CCL2 inhibitors or CCR2 antagonists has given promising results in the therapy of cancers such as breast cancer [248,249], glioma [250,251,252], and hepatocellular cancer [253] in laboratory animals inoculated with cancer cells. CCR2-targeted drugs have also been tested, for example, CCR2-targeted apoptotic peptide was used in the therapy of melanoma in laboratory animals [254]. Anti-cancer properties have also been shown in the clinical trials of carlumab (CNTO888), a monoclonal antibody against CCL2, in patients with solid tumors [255].

Another therapeutic approach is to block the CCL2→CCR2 axis in chemotherapy or radiotherapy [256]. Chemotherapy induces an increase in CCL2 expression which results in apoptosis resistance [257]. As CCL2 causes a resistance to drugs such as cabazitaxel [157], docetaxel [257], and tamoxifen [158], blocking this chemokine increases the effectiveness of chemotherapy. CCL2 expression is also elevated by radiotherapy in breast cancer [107], colon cancer [258], head and neck squamous cell carcinoma [138], and pancreatic ductal adenocarcinoma [259], which leads to the resistance of these cells to radiotherapy [138,258,259,260] and causes their migration and then metastasis [261]. In addition, a radiotherapy-induced increase in CCL2 expression leads to neuroinflammation [262,263] and vascular dysfunction which leads to impaired lung function [264]. This indicates that the effectiveness of radiotherapy may be enhanced by the use of CCL2-neutralizing antibodies or CCR2 antagonists.

On the other hand, an increased expression of CCL2 and CCL7 has shown anti-tumor effects in cervical carcinoma [150], glioma [146] and mastocytoma [149] induced by gene therapy in laboratory animals. These effects included tumor infiltration by CD4+ and CD8+ cells [148,149], and NK cells [146,148,150]. CCL2 also causes tumor infiltration by type 1 cytotoxic γδ T lymphocytes in melanoma [11] and by cytotoxic T lymphocytes in colon cancer [147]. As all the aforementioned cells have anti-cancer properties, in some tumors, gene therapy can be used to increase the expression of CCL2, which in turn will enhance the accumulation of anti-cancer immune cells in the tumor. In contrast, in some tumors, such as in non-small cell lung cancer, CCL2 participates in tumor immune evasion [265,266]. In this case, the effectiveness of immunotherapy can be increased by the use of CCL2 neutralizing antibodies.

Compared to other chemokines, there are few studies exploring the role of eotaxins (CCL11, CCL24, and CCL26) in tumors and little has been published on therapies directed against these chemokines. However, some tumors, including colorectal cancer, have been reported to show an increased expression of eotaxins [267], which indicates the potential benefits of immunotherapy in which anti-tumor cells will have an increased expression of CCR3, a receptor for eotaxins [267]. Such cells will accumulate in tumors exhibiting an increased expression of ligands for this receptor when compared to normal tissue.

Animal studies on inoculated tumors, such as bladder cancer [268] and renal cell carcinoma [269], showed the anti-tumor effects of mogamulizumab or Affi-5 (monoclonal antibodies targeting CCR4) which reduced the number of Treg and increased the number of NK cells and CD4+ T cells in the tumor. Another therapeutic approach is the transduction of CCR4 to cytotoxic T cells [270], lymphocytes which accumulate in a tumor with a high expression of CCR4 ligands, for example, in pancreatic cancer [270], where they destroy cancer cells. In some cancers, an increased expression of CCR4 ligands (CCL17 and CCL22) in a tumor may play an anti-tumor role in colon carcinoma [212], lung cancer [271], ovarian cancer [272] and melanoma [211,272], resulting in an increase in CD4+ T cells and CD8+ T cells. An increased expression of CCL17 and CCL22 is consistent with the prognosis for patients with colon, lung, and ovarian cancers—the higher the expression of these chemokines in these tumors, the better the prognosis [8,9,229].

Clinical trials were also carried out in patients with various solid tumors who were administered mogamulizumab (KW-0761) [273,274], a defucosylated humanized monoclonal antibody targeting CCR4. This therapy led to a decrease in the number of Treg in the blood and caused changes in the cancer microenvironment which facilitated the effectiveness of immunotherapy. Mogamulizumab has also shown promising results in clinical trials involving patients with adult T-cell leukemia/lymphoma [275,276], cutaneous T-cell lymphoma [277,278] and peripheral T-cell lymphoma [275,277]. The use of mogamulizumab is also postulated in the treatment of the Epstein-Barr virus (EBV)-associated T/NK-cell lymphoproliferative diseases due to the expression of CCR4 in cells infected with EBV [279].

## 7. Conclusions

CC chemokines are an important component of the tumor microenvironment. Produced by tumor cells and tumor-associated cells such as CAF, TAM, and TAN, CC chemokines increase the proliferation, migration, and invasion of cancer cells, and induce their drug resistance. If a specific CC chemokine receptor is expressed on a circulating cancer cell, it will migrate to organs showing a high expression of the ligand of that receptor. In a similar way, CC chemokines recruit tumor-associated cells into the tumor niche—this mode of action of CC chemokines is local and depends on other factors in the tumor microenvironment. Yet although the recruited tumor-associated cells can enhance the growth of a tumor, some of them (namely, TIL), do have anti-cancer properties. This shows the importance of adjusting the therapeutic approach to the specific context in which a given CC chemokine operates. Therapies should try to use the anti-cancer properties of a given chemokine or suppress its pro-cancer properties. For example, they may increase the expression of a given chemokine in a tumor and then apply immunotherapy, in which anti-cancer immune cells accumulate in the tumor via this chemokine. On the other hand, they may also concentrate on blocking chemokines responsible for tumor immune evasion, and only then use immunotherapy.

## Figures and Tables

**Figure 1 ijms-21-08412-f001:**
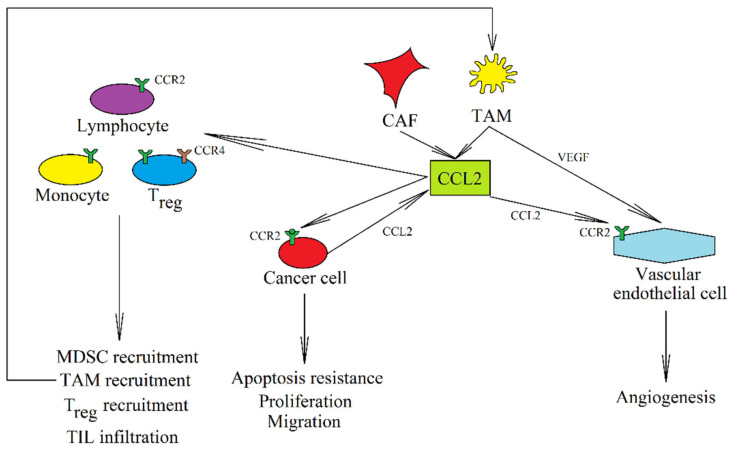
The role of CCL2 in caner. In a tumor, CCL2 is produced by tumor cells and by CAF and TAM. It activates its receptor, CCR2, on a tumor cell, which stimulates the proliferation of cancer cells and causes their migration and resistance to apoptosis. CCL2 also acts on non-cancer cells, e.g., activating CCR2 on vascular endothelial cells which results in angiogenesis. CCL2 causes the recruitment of MDSC, TAM, and T_reg_ into the tumor niche but also induces the infiltration of the tumor by anti-cancer TIL.

**Figure 2 ijms-21-08412-f002:**
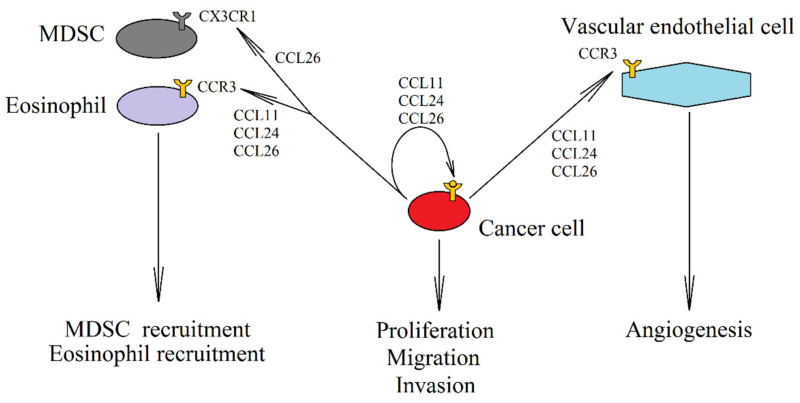
The role of eotaxins in cancer. In a neoplastic cell there is an expression of CCL11 (eotaxin-1), CCL24 (eotaxin-2) and CCL26 (eotaxin-3). This increases the autocrine proliferation and causes the migration of cancer cells with CCR3 expression. Eotaxins also activate the CCR3 receptor on endothelial cells, which results in angiogenesis. Another effect of an increased expression of eotaxins in the tumor is the recruitment of cells into the tumor niche, in particular eosinophils, by all three eotaxins through the CCR3 receptor. MDSC are recruited into the tumor by CCL26 via CX3CR1.

**Figure 3 ijms-21-08412-f003:**
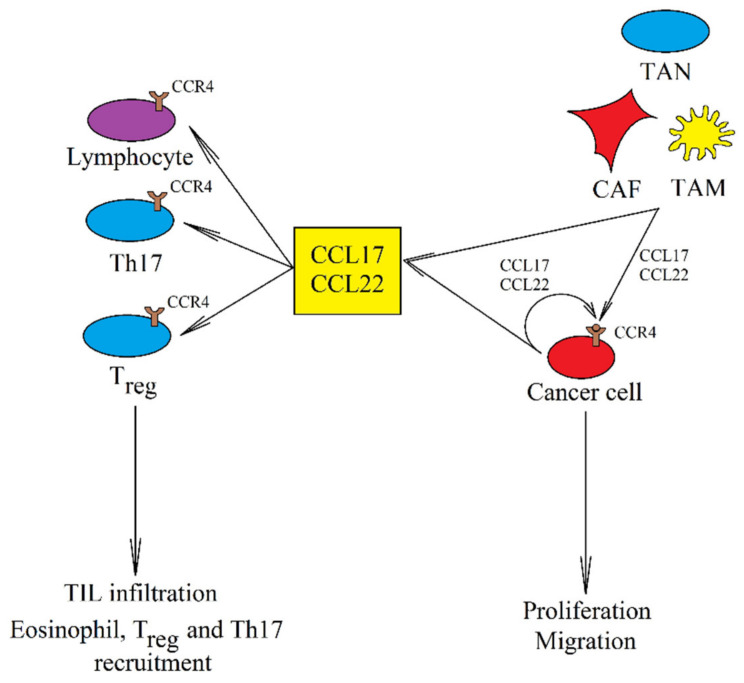
The role of CCL17 and CCL22 in cancer. These two chemokines are produced by cancer cells, CAF, TAM, and TAN. They increase the proliferation of cancer cells and their migration and invasion. They also cause the recruitment of T_reg_ and Th17. However, they can also cause infiltration of the tumor by TIL, which has an anti-cancer effect.

**Table 1 ijms-21-08412-t001:** Influence of increased expression of individual CC chemokines discussed in this review on the prognosis of patients with various cancers according to “The Human Protein Atlas” (https://www.proteinatlas.org/) [8,9].

Type of Cancer	Chemokine												
	CCL2	CCL7	CCL8	CCL11	CCL13	CCL14	CCL15	CCL16	CCL17	CCL22	CCL23	CCL24	CCL26
Glioma	↓	↓	↓*p* = 0.095	N/A	↓*p* = 0.054	--	N/A	N/A	↑	--	↓	↓	--
Thyroid cancer	--	--	↑*p* = 0.057	--	--	--	↑	--	↓	↓	↓	--	--
Lung cancer	↓*p* = 0.075	↓	--	↓*p* = 0.081	↑*p* = 0.085	--	--	--	↑	↑*p* = 0.089	--	--	↓*p* = 0.062
Colorectal cancer	↓	↓*p* = 0.085	↓	↑	↑	↑	↑	--	↑*p* = 0.058	↑	↓*p* = 0.10	↑	--
Head and neck cancer	↓*p* = 0.086	--	↑	↑*p* = 0.058	--	↑	N/A	--	↑	↑	--	↑*p* = 0.056	↓
Stomach cancer	--	--	--	↓	--	↓	--	↓	--	↑*p* = 0.077	--	↑	↑
Liver cancer	↑	N/A	--	--	↓*p* = 0.076	↑	↓*p* = 0.089	↑	--	--	↑	--	↓
Pancreatic cancer	--	↓	--	↓	↓	↑	--	↑	--	↑	↑*p* = 0.071	--	↑*p* = 0.079
Renal cancer	↓	↓	↓	↓	↓	↓	↓	↓	↓*p* = 0.060	↑	↓	--	↓
Urothelial cancer	↓	↓	↓	↓	↑	--	↑	↑	↑	↑	--	↓	↓
Prostate cancer	--	↑*p* = 0.070	--	--	↓*p* = 0.060	↑	↑	--	↓	--	↑*p* = 0.069	--	--
Testicular cancer	↓	↓*p* = 0.087	↓	↓	--	--	--	--	↓	↓	--	--	--
Breast cancer	↑*p* = 0.064	↑*p* = 0.077	↓*p* = 0.075	↑	↑	↑	--	--	↑	↑	↑	↑	--
Cervical cancer	↓	↓	--	--	↑	↑*p* = 0.058	--	--	↑	↑	↑	--	--
Endometrial cancer	↑*p* = 0.056	↓	↓	↓*p* = 0.082	↑	↓	--	↓	↑	↑	↓*p* = 0.095	↑	--
Ovarian cancer	--	↑	↑	--	↑	↓*p* = 0.081	N/A	--	↑	↑	↑	--	↑*p* = 0.087
Melanoma	--	--	--	--	--	--	--	--	↑*p* = 0.064	--	↑*p* = 0.065	↓	--

↑ blue background—better prognosis with higher expression of a given chemokine in a tumor; ↓ red background—worse prognosis with higher expression of a given chemokine in a tumor; -- means no correlation with higher expression of a given chemokine in a tumor.

**Table 2 ijms-21-08412-t002:** Effects of increased expression of individual CC chemokine receptors discussed in this review on the prognosis of patients with various cancers according to “The Human Protein Atlas” (https://www.proteinatlas.org/) [8,9].

Type of Cancer	Receptor			
	CCR1	CCR2	CCR3	CCR4
Glioma	↓*p* = 0.095	↓	--	--
Thyroid cancer	--	↑	--	--
Lung cancer	↓*p* = 0.088	↑	↓	↑
Colorectal cancer	--	↑	--	↑
Head and neck cancer	↑*p* = 0.067	↑	↓	↑
Stomach cancer	--	--	↓	--
Liver cancer	--	↑*p* = 0.081	↓	--
Pancreatic cancer	--	--	↓	--
Renal cancer	↓	↓	↓	↓
Urothelial cancer	--	--	↓*p* = 0.098	--
Prostate cancer	--	--	↑	--
Testicular cancer	↓	↓	↓	↓
Breast cancer	--	↑	↓	↑
Cervical cancer	--	↑	↓	↑
Endometrial cancer	--	↑	↓	↑
Ovarian cancer	↑	↑	↑	↑
Melanoma	↑	↑*p* = 0.063	↓	↑

↑ blue background—better prognosis with higher expression of a given chemokine in a tumor; ↓ red background—worse prognosis with higher expression of a given chemokine in a tumor; -- means no correlation with higher expression of a given chemokine in a tumor.

**Table 3 ijms-21-08412-t003:** CC chemokines discussed in this part of the article, including cells recruited to the tumor niche.

Name	Receptor	Effect on the Recruitment of Non-Cancer Cells into the Tumor	Induction of Angiogenesis or Lymphangiogenesis	Organ-Specific Metastasis
CCL2	CCR1 (low-affinity binding), CCR2, CCR3 (antagonist), CCR4, CCR5	TIL, MDSC, MSC, TAM, T_reg_, Th17, neural progenitor cells, microglia, hepatic stellate cells	Angiogenesis	Bone, perineural invasion
CCL7	CCR1, CCR2, CCR3, CCR5 (antagonist)	TIL, TAM		
CCL8	CCR1, CCR2, CCR3, CCR5	TAM, T_reg_		
CCL11	CCR2 (antagonist), CCR3, CCR5	Eosinophils	Angiogenesis	
CCL13	CCR2, CCR3, CCR5			
CCL14	CCR1, CCR3 (low-affinity binding), CCR5	TAM	Angiogenesis	
CCL15	CCR1, CCR3	MDSC, MSC, TAM, TAN, osteoclast precursors, osteoclasts	Angiogenesis	
CCL16	CCR1, CCR2, CCR5, CCR8, histamine H4 receptor		Angiogenesis	
CCL17	CCR4	TIL, T_reg_, Th17, eosinophils		
CCL22	CCR4	TIL, T_reg_, Th17, eosinophils		Bone
CCL23	CCR1		Angiogenesis	
CCL24	CCR2 (antagonist), CCR3,	Eosinophils	Angiogenesis	
CCL26	CCR1 (antagonist), CCR2 (antagonist), CCR3, CCR5 (antagonist), CX3CR1	Eosinophils, MDSC, TAM	Angiogenesis	

MDSC—myeloid-derived suppressor cells; MSC—mesenchymal stem cells; TAM—tumor-associated macrophages; TAN—tumor-associated neutrophils; Th17—T helper 17; TIL—anti-cancer tumor-infiltrating lymphocytes; T_reg_—regulatory T cells.

**Table 4 ijms-21-08412-t004:** Receptors of CC chemokines discussed in this part of the paper, with their ligands and functions in a tumor.

Receptor	Ligand	Influence on the Recruitment of Cells into the Tumor Niche	Effects on Tumor Vascularization	Organ-Specific Metastasis
CCR1	CCL2, CCL3, CCL4, CCL5, CCL7, CCL8, CCL14, CCL15, CCL16, CCL23	MDSC, MSC, TAM, TAN, osteoclast precursors, osteoclasts	Increase in VEGF expression which leads to angiogenesis	Liver
CCR2	CCL2, CCL7, CCL8, CCL13, CCL16	MDSC, MSC, TAM, T_reg_	TAM-dependent angiogenesis	Bone, perineural invasion
CCR3	CCL5, CCL7, CCL8, CCL11, CCL13, CCL14, CCL15, CCL24, CCL26, CCL28	Eosinophils, TAM	Angiogenesis	
CCR4	CCL2, CCL17, CCL22	TIL, Th17, T_reg_,		Lymph node, bone

MDSC—myeloid-derived suppressor cells; MSC—mesenchymal stem cells; TAM—tumor-associated macrophages; TAN—tumor-associated neutrophils; Th17—T helper 17; TIL—anti-cancer tumor-infiltrating lymphocytes; T_reg_—regulatory T cells; VEGF—vascular endothelial growth factor.

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
