# Peer review of "CC Chemokines in a Tumor: A Review of Pro-Cancer and Anti-Cancer Properties of the Ligands of Receptors CCR1, CCR2, CCR3, and CCR4"

_ijms, 2020, doi:10.3390/ijms21218412_

Round 1
Reviewer 1 Report
For a clinician who treat cancer the most important part of the manuscript was pa paragraph 6. CC chemokines in therapy, however this section is short and dull. For my opinion it should be elaborated and should become the peak of the manuscript. Theory is nice and important however we would like to know how to utilize it to save patients life.
Author Response
Review 1
For a clinician who treat cancer the most important part of the manuscript was pa paragraph 6. CC chemokines in therapy, however this section is short and dull. For my opinion it should be elaborated and should become the peak of the manuscript. Theory is nice and important however we would like to know how to utilize it to save patients life.
The section about therapy has been significantly extended.

Reviewer 2 Report
The present manuscript is a review paper about the CC chemokines in tumors. The topic is timely, even if not very novel. The authors systematically reviewed the latest progress in the physiological functions and roles of the CC chemokines, which are the ligands of four receptors (CCR1, CCR2, CCR3, and CCR4, etc.) in tumor initiation, progression, and metastasis, as well as their functions and roles in angiogenesis and lymph-angiogenesis. In addition, they also discussed the effects of each chemokine on the recruitment of cancer-associated cells to the tumor niche, and also the recruitment of tumor-infiltrating lymphocytes (TIL), which exhibits the anticancer properties of CC chemokines. The authors collected huge information on these receptors and related chemokine ligands. The manuscript is overall clearly written, easy to understand. It would be better to give a final conclusion about the functions and roles of these CC chemokines in the abstract and in the main text.
The below comments should be addressed to strengthen this manuscript before its publication.
Major comments:
- This article would benefit from a close editing.
- The title looks strange, should it be “CC chemokines in tumor (or in tumors). A review of pro-cancer and anti-cancer properties of the ligands for the receptors CCR1, CCR2, CCR3, and CCR4”?
- In abstract, a conclusion statement should be given, not just present the others’ works.
- Through the manuscript, the authors mention several times of “cancer processes”, what does that mean? This saying is not clear. The chemokines play different roles either in cancer initiation, or progression, or metastasis. It’s better to clear the function or the role of that chemokine plays in which stage.
Minor comments:
- In table 3: The distance between the columns is too small, it’s difficult to understand what the title line of the table is discussing.
Author Response
Review 2
The present manuscript is a review paper about the CC chemokines in tumors. The topic is timely, even if not very novel. The authors systematically reviewed the latest progress in the physiological functions and roles of the CC chemokines, which are the ligands of four receptors (CCR1, CCR2, CCR3, and CCR4, etc.) in tumor initiation, progression, and metastasis, as well as their functions and roles in angiogenesis and lymph-angiogenesis. In addition, they also discussed the effects of each chemokine on the recruitment of cancer-associated cells to the tumor niche, and also the recruitment of tumor-infiltrating lymphocytes (TIL), which exhibits the anticancer properties of CC chemokines. The authors collected huge information on these receptors and related chemokine ligands. The manuscript is overall clearly written, easy to understand. It would be better to give a final conclusion about the functions and roles of these CC chemokines in the abstract and in the main text.
The below comments should be addressed to strengthen this manuscript before its publication.
Major comments:
- This article would benefit from a close editing.
- The title looks strange, should it be “CC chemokines in tumor (or in tumors). A review of pro-cancer and anti-cancer properties of the ligands for the receptors CCR1, CCR2, CCR3, and CCR4”?
The title has been changed.
- In abstract, a conclusion statement should be given, not just present the others’ works.
The main conclusion is now included in the abstract, mentioning that chemokines have both pro- and anti-cancer properties. However, we have added a short section presenting the conclusions and clinical implications
- Through the manuscript, the authors mention several times of “cancer processes”, what does that mean? This saying is not clear. The chemokines play different roles either in cancer initiation, or progression, or metastasis. It’s better to clear the function or the role of that chemokine plays in which stage.
Hanahan&Weinberg give a couple of "hallmarks of cancer", including proliferation, metastasis migration, tumor immune evasion, and angiogenesis. In this paper we show the significance of CC chemokines for these "hallmarks of cancer"and in the text we have made some changes so that there are no misunderstandings.
In this work we showed in detail the importance of individual CC chemokines in metastasis and progression. The influence of chemokines in cancer initiation is poorly researched. However, we tried to show the properties of each of the CC chemokines, which allows to easily draw conclusions about their role in cancer initiation.
Minor comments:
- In table 3: The distance between the columns is too small, it’s difficult to understand what the title line of the table is discussing.
The lines separating individual columns in Tables 3 and 4 have been added. There should be no more problems with clarity.
